# Tourism Stakeholder Perspectives on Corporate Social Responsibility in Serbia: The Perception of Hotel Employees

Maja Mijatov Ladičorbić [1], Aleksandra S. Dragin [1,2], Tamara Jovanović [1], Milica Solarević [1], Olja Munitlak Ivanović [1], Vladimir Stojanović [1,2], Kristina Košić [1], Anđelija Ivkov Džigurski [1], Slavica Tomić [3], Miroslav D. Vujičić [1], Milena Nedeljković Knežević [1], Ivana Blešić [1,4], Željko Anđelković [5,*], Zrinka Zadel [6], Jelena Tepavčević [1] and Aleksandra Stoiljković [3]

1    Faculty of Sciences, Department of Geography, Tourism and Hotel Management, University of Novi Sad, 21000 Novi Sad, Serbia
2    Novi Sad School of Business, 21000 Novi Sad, Serbia
3    Faculty of Economics in Subotica, University of Novi Sad, 24000 Subotica, Serbia
4    Institute of Sports, Tourism and Service, South Ural State University, 76 Lenin Ave., Chelyabinsk 454080, Russia
5    National Museum Niš, 18000 Niš, Serbia
6    Faculty of Tourism and Hospitality Management, University of Rijeka, 51000 Rijeka, Croatia
*    Correspondence: zeljkozelja@yahoo.co.uk

**Abstract:** Individual values shaped in the early years of each individual might be reflected in the perception of the business environment. Therefore, this research explored whether there are significant differences in employees' perceptions of the established dimensions of corporate social responsibility (CSR), namely philanthropic, legal, economic and ethical, based on differences in the importance of individual values (IV). The research results show that each of the CSR dimensions differs according to the respondents' IVs. More precisely, there are significant differences in the respondents' perceptions of the philanthropic dimension of CSR based on differences in the evaluation of sense of belonging, warm relationships, being well respected, fun and enjoyment of life, security, self-respect and sense of accomplishment. Differences in perceptions of the legal dimension of CSR exist only in the case of differences in the evaluation of self-respect. The research results also show that there are significant differences in perceptions of the economic dimension of CSR, based on differences in the evaluation of sense of belonging, warm relationships, fun and enjoyment in life, security, self-respect and sense of accomplishment. Finally, significant differences in the perception of the ethical dimension of CSR exist in the case of differences in sense of belonging, excitement, warm relationships, being well respected, fun and enjoyment of life, security, self-respect and sense of accomplishment. The results could provide the basis of information on how IVs can motivate employees to understand and participate in the proactive management of CSR activities in sensitive environments, such as national parks or other protected areas that become a central meeting place for tourists and employees.

**Keywords:** corporate social responsibility; individual values; hotel employees' perception; developing country; employee behavior

## 1. Introduction

Corporate social responsibility (CSR) is widely used but at the same time inconsistent in its application, which has led to different definitions of this term [1,2]. In an attempt to define CSR as precisely as possible, it has been broadly defined, e.g., as behavior that involves social improvement beyond a company's interest, to a narrower view that focuses on economic benefits to a company's shareholders [3,4]. CSR has traditionally been conceptualized more as "the commitment of managers to take action to protect and improve both the welfare of society as a whole and the interests of organizations" [5]. Furthermore, CSR is about "bringing corporate behavior to a level consistent with currently prevailing societal

norms, values, and performance expectations" [6]. The socially responsible company must seek to make profits while respecting the laws, ethical code and demands of local people and the community. In other words, CSR goes beyond the legal and economic framework to include responsibility not only for the preservation of the natural environment, but also for a variety of stakeholders that include the local population. There is no question that the population and the local community contribute to realistic planning and investment in the management system and find solutions that are affordable and suitable for citizens [7,8]. Regardless of some differences in definition, it is significant that CSR is a multidimensional construct [9,10]. It also influences numerous stakeholders. "CSR is, at its core, a multi-theoretic and multi-stakeholder construct that requires examination based on the context of the study" [11].

Following the wider concept of CSR, Carroll [12] argued that CSR needs to include four types of responsibilities: economic, legal, ethical and philanthropic. His categorization was widely accepted and applied in numerous research [13–17]. Economic responsibilities require businesses to be profitable and produce goods and services which are desirable in society. Legal responsibility corresponds to society's expectation that companies fulfil their economic obligations within the framework of the law. Ethical responsibility requires that companies follow the behaviors that are considered morally right [18]. Finally, philanthropic responsibilities reflect the common desire to see businesses become actively involved in the betterment of society beyond their economic, legal and ethical responsibilities [19]. Carroll's concept of CSR (1979) was the starting point of this research. As of November 2022, Carroll's research has been cited, according to Google Scholar, in over 14 thousand articles from different scientific fields, and Carroll's total scientific opus is cited in more than 60 thousand scientific papers published worldwide.

Although the concept of CSR has been applied to a wide range of industries, previous research has shown that CSR practices can vary significantly within industries [20]. Song and Wen [21] also suggested that behavioral patterns may be similar within an industry but differ across industries. Therefore, they proposed to investigate how CSR is influenced in different industries. This research focuses on the hotel industry in Serbia, as an example of developing countries. Employees' perceptions of CSR were mostly researched in the context of their effects on organizational internal outcomes, such as employees' performances and employees' corporate identification, as well as with employees' perception of quality of work life and their work motivation patterns [22]. Although the perceptions of employees were previously researched in terms of the national culture [23], this research is limited in terms of relations with employees' individual values. In this regard, this research focuses on differences in perceptions of socially responsible behavior based on differences in individual values of employees in hotels in Kopaonik, a well-known winter tourism destination and declared national park in the country, which raises expectations of corporate social responsibility in such an environment. Zhang et al. [24] indicated that research on the micro-level consequences (individual outcomes) of CSR is still limited. Concerning that, the main goal of this research is to explore whether there are significant differences in the employees' perception of established dimensions of corporate social responsibility, based on differences in the importance of individual values. According to the study conducted by Wong et al. [25], previous studies regarding CSR were more recently conducted in well-developed territories. The contribution of this research is reflected in the fact that it is one of the first studies aimed at examining this important topic within the business environment of dominating small and medium hotels in a transitional country. Finally, as Madanaguli et al. [11] indicated, aside from CSR being widely researched within the tourism literature, these reviews are often characterized by a narrow scope and without a comprehensive understanding of CSR. More precisely, these authors highlighted that intervening variables, which could provide a detailed understanding of individual differences that are shaping the CSR initiatives, such as personality dimensions, are never studied [11].

## 2. Literature Review

Falk and Heblich [26] as well as Zhang and Ouyang [4] suggested that a company's goal of surviving in the marketplace and thriving commercially is best achieved through long-term planning, acceptance and understanding that if it treats society well, society may reciprocate in kind. Post, Preston and Sachs [27] and Wang, Hu and Zhang [28] went even further and argued that a company cannot and should not do business if it does not take responsibility for all aspects of business, including the well-being of the population and the wider society. All of this suggests that businesses should be aligned with a range of social norms and standards [29]. Therefore, CSR plans have to be pragmatic in achieving what is suitable for a specific local community and its issues, with the necessary involvement of state authorities [28,30].

Furthermore, scholars have suggested that corporations engage in socially responsible activities for a variety of reasons, including a positive image and increased likelihood of hiring top-quality employees, as well as a positive impact on the company's bottom line and the environment when they are operating [31–33]. The marketing literature often focuses on the role of marketing in managing stakeholder (especially customer) perceptions and the impact of CSR on the (corporate) brand. Carrying out good deeds generates a positive public relations story [34]. The same authors find that while CSR is most commonly explained in terms of the strategic commercial interest of the organization (image and reputation management, the manipulation of stakeholder manipulation and integration of the organization into its host community), this is not always the case. CSR initiatives can lie on a motivational axis in terms of strategic or moral drivers and also on an axis labeled as the locus of responsibility. When individual managers have the power to make decisions, they may start or change certain projects to address their moral concerns. Evidence that individual managers practice social responsibility, as opposed to merely serving as agents of corporate policy, may therefore encourage or serve as a reminder to staff members that they can "make the difference" in an organization even in the absence of a formalized CSR culture. This is particularly crucial in sensitive environments, such as protected areas [33,34].

### 2.1. Corporate Social Responsibility in the Hotel Industry in Serbia

Hotels affiliated with international hotel chains have been pioneers in implementing the concept of corporate social responsibility in the hotel industry in Serbia. These hotels have the necessary infrastructure in terms of modern technologies, capital, and knowledge to carry out organizational social responsibility. In recent years, hotel chain branches in Serbia have implemented a variety of social responsibility programs to assist in the resolution of environmental and community issues. In addition to addressing significant social issues, these hotels have improved their relationships with customers, employees and suppliers [12,35].

However, domestic hotels in Serbia, which mostly belong to small and medium enterprises and dominate in international hotel chains, have far fewer capabilities, so the implementation of socially responsible business is still at an initial level, which is in line with the results of previous studies conducted in hotels in Serbia [36–40]. However, hoteliers who implement socially responsible activities in Serbia point out that they are satisfied with the results of the implementation of these programs which can provide numerous benefits in terms of cost reduction, quality increase, working atmosphere, customer relations, business partners and community. This is reflected in how the hotel is perceived, as well as in the stakeholders' loyalty and trust. Milovanović [41] points out that an increase in the implementation level of social responsibility in hotels in Serbia can be expected, according to the ratio of investments in such programs and their benefits, as well as the increasing number of foreign tourists in Serbia and their awareness of social responsibility. As a result, social responsibility can be a powerful tool for market differentiation and gaining a competitive advantage, particularly in times of economic crisis and globalization [41].

## 2.2. Individual Values

Previous studies suggest that employees' personal characteristics may also affect attitudes toward CSR [16,42–44]. These characteristics include gender, age, education, length of work experience, managerial experience and international work experience [16,42,43]. Psychological rewards, a sense of achievement, recognition of learning opportunities and personal self-realization are just some of the aspects of individual motivation of employee participation in the acceptance and implementation of socially responsible activities. An increasing number of companies are considering such practices [45] and are becoming aware that supporting employees' individual values can help motivate them to understand and participate in the proactive management of socially responsible activities [46]. However, for now, organizational characteristics represent a more complex area of research compared to individual values, regarding their impact on the perception of socially responsible activities [47–49].

According to McCrimmon [50], the values that individuals find more difficult to shape at a young age, and that, once established, reflect on behavior and perception in the business environment, should be considered. Furthermore, their importance may change at different stages of life, depending on the circumstances, environment, and priorities that are important to the individual at the time [51]. In this research, the List of Individual Values developed by Kahle [52] was applied. This list includes nine different individual values (IVs), represented in the form of feelings of belonging, excitement, intimacy with others, self-realization, respect, fun and enjoyment of life, security, self-respect and a sense of achievement. The responses were collected using a five-point Likert scale. The research aimed to determine whether there are statistically significant differences between the importance of the level of individual values in the everyday life of employees of hotels in the National Park of Kopaonik and their perception of the dimensions of social responsibility for hotels in which they are employed.

## 3. Research Methodology

### 3.1. Instrument

The research was conducted by using a questionnaire, which consisted of questions related to *socio-demographic characteristics* of employees in hotels in Kopaonik (such as gender, age, education degree and the place of residence) and questions about two main constructs of this research—CSR and IVs.

*CSR* was examined based on the standardized and globally accepted Model of Social Performances [29,53–56]. The questionnaire, based on this model, contains items regarding four dimensions of CSR. The first dimension refers to the perception of employees on how many hotels participate in charitable activities and care for the local community. This dimension is marked as the philanthropic dimension of CSR. The second dimension is marked as the legal dimension of CSR and it indicates the perception of employees on the readiness of hotels to comply with business regulations. It is about the readiness of these hotels to operate under legal contracts and other labor acts. The third dimension refers to the perception of employees on the economic aspects of the hotel business (covering operating costs, striving to improve employee productivity and establishing a long-term strategy for economic growth of the organization, improving product quality) and it is marked as the economic dimension of CSR. The last dimension, marked as the ethical dimension of CSR, concerns the perception of employees on the ethics of the organization's behavior towards different stakeholders, which includes the relations towards employees in the organization [16].

On the other hand, *IVs* were examined based on the already mentioned standardized list of values developed by Kahle, containing the following: a sense of belonging, excitement, warm relationships, self-fulfillment, being well respected, fun and enjoyment in life, security, self-respect and a sense of accomplishment [52].

### 3.2. Data Collection Procedure and Statistical Analysis

The research results are part of a broader study on business ethics and CSR. Data collection for this study lasted from 2013 to 2018, with the main part of the research in 2013 related to establishing contacts with managers in hotels who later helped with data collection. It lasted until 2018 when the majority of answers for the final construction of CSR dimensions was gathered. Since 2018, regulations concerning CSR have not been established and there have not been significant changes at the national level. The dimensions of the CSR in hotels in Kopaonik were identified through factor analysis based on the collected data. The identified CSR dimensions are used as a starting point for conducting further analyses for this research. Therefore, within this study, it was examined whether there is a significant difference in the perception of measured dimensions of CSR, based on differences in the importance of IVs in the daily life of the respondents. Respondents answered the questions by using the 5-point Likert scale, while the main research results were gained by using the factor analysis and one-factor univariate analysis (ANOVA).

### 3.3. Sample

The research included 211 respondents. According to the research results, represented within the Table 1, the share of both genders is approximately equal, with a slightly higher share of males among the employees in hotels within Kopaonik (52.6%). Regarding the age structure, respondents aged between 21 and 30 years (49.8%) predominate, while the share of respondents aged between 31 and 40 years (21.8%) is also significant. The research results pointed to an approximate share of the respondents up to 20 years (10.9%) and respondents aged between 41 and 50 years (10.4%). The lowest share of the respondents (7.1%) is aged between 51 and 60 years. The most common level of education among the respondents is high school (63.5%). This is followed by respondents with a university degree (two-year degree) (15.2%) and a university degree (four-year degree) (12.3%). The smallest percentage of respondents have an elementary school degree (5.7%), a master's degree (2.8%) and a doctorate (0.5%). It is important to mention that respondents come from different local municipalities in Serbia including Raška, Brus, Novi Pazar, Prijepolje, Ivanjica, Kraljevo, Čačak, Kragujevac, Kruševac, Niš, Valjevo, Belgrade, Novi Sad, Zrenjanin and Vrbas. Represented socio-demographic profile of the sample by any segment corresponds to the general structure of the employees in hotels, especially in facilities with seasonal oscillations of work, such as hotels in Kopaonik.

**Table 1.** Socio-demographic characteristics of the respondents.

| *Gender* | | *Education Degree* | |
|---|---|---|---|
| Males | 52.6% | Primary education | 5.7% |
| Females | 47.4% | High school | 63.5% |
| *Age* | | College degree (two-years study) | 15.2% |
| Up to 20 | 10.9% | University degree (four-years study) | 12.3% |
| 21–30 | 49.8% | Master's degree | 2.8% |
| 31–40 | 21.8% | PhD | 0.5% |
| 41–50 | 10.4% | | |
| 51–60 | 7.1% | | |

Source: research results.

## 4. Results

The analysis of the main components of corporate social responsibility obtained 17 items. The value of Kaiser–Meyer–Olkin's indicator is 0.905, which is exceeding the recommended value of 0.6. Bartlett's test for sphericity reached statistical significance ($p = 0.000$). Thus, the use of factor analysis was justified. Principal component analysis revealed the presence of four components with values over 1, which explained 41.956%, 10.889%, 6.729% and 6.223% of the variance. After the factor extraction, their rotation was performed by using the Promax method. The result was four factors, namely philanthropic

dimension of CSR, legal dimension of CSR, economic dimension of CSR, and ethical dimension of CSR (the detailed construction of each dimension is shown in Table 2).

**Table 2.** Factor analysis of CSR (Promax rotation).

| Scale ($\alpha$ = 0.917) Model of Social Performances (Carroll, 1979) | Philanthropic Dimension of CSR ($\alpha$ = 0.883) | Legal Dimension of CSR ($\alpha$ = 0.909) | Economic Dimension of CSR ($\alpha$ = 0.880) | Ethical Dimension of CSR ($\alpha$ = 0.824) |
|---|---|---|---|---|
| My organization is participating in campaigns oriented towards providing help to people in trouble. | 0.950 | | | |
| My organization is trying to help the community. | 0.897 | | | |
| My organization is participating in voluntary activities. | 0.850 | | | |
| My organization provides various donations. | 0.836 | | | |
| Work in Kopaonik is intensive. | 0.367 | | | |
| My organization operates in accordance with the labor law acts. | | 0.957 | | |
| My organization is committed to doing business in accordance with legal contracts. | | 0.891 | | |
| My organization adheres business regulations. | | 0.771 | | |
| My organization contains established rules and methods of working with the costumers. | | 0.768 | | |
| My organization strives towards covering the operating costs. | | | 0.832 | |
| My organization strives towards establishing the long-term strategy for providing the economic growth. | | | 0.761 | |
| My organization generates the impact of employment. | | | 0.758 | |
| My organization is striving towards improving the employees' productivity. | | | 0.707 | |
| My organization protects its employees in terms of too demanding customers. | | | | 0.838 |
| Employees are open for cooperation. | | | | 0.708 |
| My organization provides adequate living conditions for seasonal workers. | | | | 0.664 |
| My organization encourages openness and cooperation between colleagues. | | | | 0.660 |

Source: research results.

In terms of the respondents' ICs, the research results showed that the IVs of self-respect and being well respected are often very important in the daily life of the respondents (M = 4.34). Approximate values were also recorded for security (M = 4.28), sense of accomplishment (M = 4.27) and self-fulfillment (M = 4.18), as well as the importance of establishing the warm relationships (M = 4.17). Slightly lower mean values were recorded for the importance of fun and enjoyment in life (M = 3.96) and sense of belonging (M = 3.95), while the lowest mean value was recorded for excitement (M = 3.58), which might be seen in Table 3.

**Table 3.** Individual values according to Kahle's list of values (1983) scale.

| Individual Value List of Values (Kahle, 1983) | Min | Max | MV | SD |
|---|---|---|---|---|
| Sense of belonging | 1 | 5 | 3.95 | 0.96221 |
| Excitement | 1 | 5 | 3.58 | 1.17404 |
| Warm relationships | 1 | 5 | 4.17 | 0.88693 |
| Self-fulfillment | 1 | 5 | 4.18 | 1.02007 |
| Being well respected | 1 | 5 | 4.34 | 0.92949 |
| Fun and enjoyment in life | 1 | 5 | 3.96 | 1.15184 |
| Security | 1 | 5 | 4.28 | 0.98788 |
| Self-respect | 1 | 5 | 4.34 | 0.91919 |
| Sense of accomplishment | 1 | 5 | 4.27 | 0.95901 |

Source: research results.

Results of one-factor univariate analysis indicated that there are significant differences in the employees' perception of philanthropic dimension of CSR on the basis of sense of belonging (F = 5.441, $p$ = 0.000), warm relationships (F = 6.918, $p$ = 0.000), being well respected (F = 5.461, $p$ = 0.000), fun and enjoyment in life (F = 2.701, $p$ = 0.032), security (F = 6.629, $p$ = 0.000), self-respect (F = 5.108, $p$ = 0.001) and sense of accomplishment (F = 6.047, $p$ = 0.000). Results of the ANOVA test of perception for the philanthropic dimension of CSR according to the respondents' IVs are represented within Table 4, by representing concrete statistical results for each observed group. It could be seen that the perception of the philanthropic dimension of CSR differs significantly between the respondents who indicated that a sense of belonging is rarely important in their everyday life (M = 2.31) and those who indicated that this IV is always important (M = 3.56). In addition, it could be seen that the perception of the philanthropic dimension of CSR for the respondents who indicated that sense of belonging is always important differs significantly from the perception of the respondents who indicated that sense of belonging is sometimes important (M = 2.93).

A more pronounced perception of the philanthropic dimension of CSR is shown by respondents who indicated that building warm relationships is always important (M = 3.54). A significant difference exists between the aforementioned group of respondents and those who indicated that building warm relationships is only sometimes (M = 2.80) or often (M = 3.01) important. Based on the importance of the following IV, which is referred to as being respected, there are also differences in the perception of the philanthropic dimension of CSR. More specifically, the research results presented in Table 4 show that the stated dimension of CSR is stronger among respondents who indicated that this IV is always important in their daily lives (M = 3.41) than among respondents who indicated that this IV is only sometimes important (M = 2.63). A similar situation was observed in the case of the fun and enjoyment of life IV. Thus, the perception of the philanthropic dimension of CSR is more pronounced among the respondents who indicated that fun and enjoyment of life are always important (M = 3.38), compared to the respondents who indicated that this IV is important only sometimes (M = 2.78).

Significant differences in the perception of the philanthropic dimension of CSR also exist in the case of the importance of security. Thus, a more pronounced perception of the philanthropic dimension of CSR is shown by the respondents who indicated that security is always important (M = 3.48), compared to the respondents who indicated that security is important rarely (M = 2.44) or only sometimes (M = 2.64). IV related to a sense of self-respect also indicated significant differences in the perception of the philanthropic dimension of CSR. Based on the research results, represented in Table 4, it could be seen that the perception of the philanthropic dimension of CSR is more pronounced among the respondents who indicated that self-respect is always important (M = 3.40), compared to the respondents who indicated that this IV is sometimes important (M = 2.48).

**Table 4.** Results of ANOVA test of perception for philanthropic dimension of CSR according to the respondents' IVs.

| Group 1 | Group 2 | Difference | Significance |
|---|---|---|---|
| Sense of belonging (F = 5.441, *p* = 0.000) | | | |
| Rarely (M = 2.31) | Always (M = 3.56) | (-) 1.25121 | 0.001 |
| Sometimes (M = 2.93) | Always (M = 3.56) | (-) 0.62813 | 0.010 |
| Warm relationships (F = 6.918, *p* = 0.000) | | | |
| Sometimes (M = 2.80) | Always (M = 3.54) | (-) 0.74349 | 0.001 |
| Often (M = 3.01) | Always (M = 3.54) | (-) 0.52645 | 0.011 |
| Being well respected (F = 5.461, *p* = 0.000) | | | |
| Sometimes (M = 2.63) | Always (M = 3.41) | (-) 0.78699 | 0.002 |
| Fun and enjoyment in life (F = 2.701, *p* = 0.032) | | | |
| Sometimes (M = 2.78) | Always (M = 3.38) | (-) 0.59167 | 0.046 |
| Security (F = 6.629, *p* = 0.000) | | | |
| Rarely (M = 2.44) | Always (M = 3.48) | (-) 1.03821 | 0.015 |
| Sometimes (M = 2.64) | Always (M = 3.48) | (-) 0.83291 | 0.004 |
| Self-respect (F = 5.108, *p* = 0.001) | | | |
| Sometimes (M = 2.48) | Always (M = 3.40) | (-) 0.91832 | 0.003 |
| Sense of accomplishment (F = 6.047, *p* = 0.000) | | | |
| Never (M = 1.60) | Sometimes (M = 3.18) | (-) 1.58065 | 0.038 |
| Never (M = 1.60) | Always (M = 3.39) | (-) 1.78761 | 0.008 |
| Rarely (M = 1.94) | Sometimes (M = 3.18) | (-) 1.23779 | 0.039 |
| Rarely (M = 1.94) | Always (M = 3.39) | (-) 1.44475 | 0.004 |

Source: research results. Note: white columns below the table fields *Group 1* and the *Group 2* represent the levels of importance of concrete individual values (represented in gray rows above them). Comparisons were made between first two white rows below stated individual values, based on the mean values represented in brackets. Detailed explanation of represented results is provided in the text.

The most significant differences were recorded in the case of the perception of the philanthropic dimension of CSR based on the IV termed a sense of accomplishment. The research results showed that the perception of the philanthropic dimension of CSR is more pronounced among the respondents who indicated that a sense of accomplishment is important always (M = 3.39) or sometimes (M = 3.18), compared to the respondents who indicated that this IV is rarely (M = 1.94) or never (M = 1.60) important.

Opposite to the perception of the philanthropic dimension of CSR, which differs according to even seven out of nine IVs, in the case of the perception of the legal dimension of CSR, results of the one-factor univariate analysis showed slightly different results. The results of the ANOVA test for the perception of the legal dimension of CSR as a function of the IVs of the respondents are shown in Table 5 by presenting concrete statistical results for each observed group. More precisely, differences in the perception of the legal dimension of CSR exist only in the case of differences in the evaluation of the self-respect IV (F = 3.252, *p* = 0.013). Based on the research results represented in Table 5, it could be seen that the perception of the legal dimension of CSR is more pronounced among the respondents who indicated that IV self-respect is often important (M = 4.22), compared to the respondents who indicated that self-respect is sometimes important (M = 3.75).

**Table 5.** Results of ANOVA test of perception for legal dimension of CSR according to the respondents' IVs.

| Group 1 | Group 2 | Difference | Significance |
|---|---|---|---|
| Self-respect (F = 3.252, *p* = 0.013) | | | |
| Sometimes (M = 3.75) | Often (M = 4.22) | (-) 0.64167 | 0.021 |

Source: research results. Note: white columns below the table fields Group 1 and the Group 2 represent the levels of importance of concrete individual values (represented in gray rows above them). Comparisons were made between first two white rows below stated individual values, based on the mean values represented in brackets. Detailed explanation of represented results is provided in the text.

The research results also showed that there are significant differences in the perception of the economic dimension of CSR, based on six out of nine examined IVs. These are IVs such as sense of belonging (F = 6.453, *p* = 0.000), warm relationships (F = 3.542, *p* = 0.016), fun and enjoyment in life (F = 2.861, *p* = 0.025), security (F = 7.373, *p* = 0.000), self-respect (F = 9.215, *p* = 0.000) and sense of accomplishment (F = 9.932, *p* = 0.000). Results of the ANOVA test of perception for the economic dimension of CSR according to the respondents' IVs are represented within Table 6, by representing concrete statistical results for each observed group.

The research results, represented in Table 6, showed that the perception of economic dimension of CSR differs significantly based on the sense of belonging IV. It could be noticed that employees' perception of economic dimension of CSR is more pronounced among the respondents who indicated that sense of belonging is important always (M = 4.07) or often (M = 3.92), compared to the respondents who indicated that this IV is important sometimes (M = 3.36) or rarely (M = 3.08). There is also a significant difference in the perception of the economic dimension of CSR based on IV, which is referred to as warm relationships. The research results indicated that perception of this dimension of CSR is more pronounced in the case of the respondents who indicated that warm relationships are always important (M = 3.98), compared to the respondents who indicated that this IV is only sometimes important (M = 3.51).

A similar situation was observed in the perception of the economic dimension of CSR due to differences in the importance of fun and enjoyment in life. More specifically, the study's findings revealed that employees' perceptions of the economic dimension of CSR are more pronounced among respondents who stated that a sense of fun and enjoyment in life is always important (M = 3.91), compared to the respondents who indicated that this IV is never important (M = 3.02). Numerous significant differences in the perception of the economic dimension of CSR could be noticed based on differences in the importance of IV termed security. Perceptions of this dimension of CSR are more pronounced among respondents who indicated that a sense of security is always (M = 3.91) or often (M = 3.91) important than among respondents who indicated that this IV is only sometimes (M = 3.20) or never (M = 2.00) important.

Significant differences also exist in employees' perceptions of the economic dimension of CSR based on IV self-respect. More specifically, the perception of the economic dimension of CSR is more pronounced among respondents who indicated that self-respect is often (M = 3.99) or always (M = 3.90) important, compared to respondents who indicated that this IV is sometimes (M = 3.03), rarely (M = 3.00) or never (M = 1.92) important. Perception of the economic dimension of CSR differs significantly even in the case of differences in the importance of a sense of accomplishment. Thus, the research results represented in Table 6 showed that the perception of the economic dimension of CSR is more pronounced among the respondents who indicated that a sense of accomplishment is important always (M = 3.98), often (M = 3.90) or sometimes (M = 3.19) in comparison with the respondents who indicated that this IV is never important (M = 1.81).

**Table 6.** Results of ANOVA test of perception for economic dimension of CSR according to the respondents' IVs.

| Group 1 | Group 2 | Difference | Significance |
|---|---|---|---|
| Sense of belonging (F = 6.453, *p* = 0.000) | | | |
| Rarely (M = 3.08) | Often (M = 3.92) | (-) 0.84209 | 0.030 |
| Rarely (M = 3.08) | Always (M = 0.4.07) | (-) 0.99499 | 0.006 |
| Sometimes (M = 3.36) | Often (M = 3.92) | (-) 0.55844 | 0.013 |
| Sometimes (M = 3.36) | Always (M = 4.07) | (-) 0.71134 | 0.001 |
| Warm relationships (F = 3.542, *p* = 0.016) | | | |
| Sometimes (M = 3.51) | Always (M = 3.98) | (-) 0.47022 | 0.045 |
| Fun and enjoyment in life (F = 2.861, *p* = 0.025) | | | |
| Never (M = 3.02) | Always (M = 3.91) | (-) 0.88636 | 0.041 |
| Security (F = 7.373, *p* = 0.000) | | | |
| Never (M = 2.00) | Often (M = 3.95) | (-) 1.95370 | 0.001 |
| Never (M = 2.00) | Always (M = 3.91) | (-) 1.91314 | 0.001 |
| Sometimes (M = 3.20) | Often (M = 3.95) | (-) 0.75579 | 0.011 |
| Sometimes (M = 3.20) | Always (M = 3.91) | (-).71522 | 0.007 |
| Self-respect (F = 9.215, *p* = 0.000) | | | |
| Never (M = 1.92) | Often (M = 3.99) | (-) 2.07500 | 0.002 |
| Never (M = 1.92) | Always (M = 3.90) | (-) 1.98669 | 0.003 |
| Rarely (M = 3.00) | Often (M = 3.99) | (-) 0.99167 | 0.026 |
| Rarely (M = 3.00) | Always (M = 3.90) | (-) 0.90336 | 0.042 |
| Sometimes (M = 3.03) | Often (M = 3.99) | (-) 0.96667 | 0.001 |
| Sometimes (M = 3.03) | Always (M = 3.90) | (-) 0.87836 | 0.001 |
| Sense of accomplishment (F = 9.932, *p* = 0.000) | | | |
| Never (M = 1.81) | Sometimes (M = 3.19) | (-) 1.37298 | 0.044 |
| Never (M = 1.81) | Often (M = 3.90) | (-) 2.08929 | 0.000 |
| Never (M = 1.81) | Always (M = 3.98) | (-) 2.16980 | 0.000 |
| Sometimes (M = 3.19) | Often (M = 3.90) | (-) 0.71630 | 0.006 |
| Sometimes (M = 3.19) | Always (M = 3.98) | (-) 0.79682 | 0.000 |

Source: research results. Note: white columns below the table fields *Group 1* and the *Group 2* represent the levels of importance of concrete individual values (represented in gray rows above them). Comparisons were made between first two white rows below stated individual values, based on the mean values represented in brackets. Detailed explanation of represented results is provided in the text.

Finally, the research results of the one-factor univariate analysis also showed that there are significant differences in the perception of the ethical dimension of CSR, based on differences in IVs. Results of the ANOVA test of perception for the ethical dimension of CSR according to the respondents' IVs are represented within Table 7, by representing concrete statistical results for each observed group. Based on the research results, represented within the Table 7, it could be seen that significant differences in the perception of the ethical dimension of CSR exist in the case of differences in eight out of nine IVs: sense of belonging (F = 6.844, *p* = 0.000), excitement (F = 3.157, *p* = 0.015), warm relationships (F = 6.146, *p* = 0.001), being well respected (F = 5.746, *p* = 0.000), fun and enjoyment in life (F = 6.481, *p* = 0.000), security (F = 6.848, *p* = 0.000), self-respect (F = 4.320, *p* = 0.002) and sense of accomplishment (F = 3.150, *p* = 0.015). Only the IV of self-fulfillment did not show a significant difference in the perception of the ethical dimension of CSR. Based on the research results, represented in Table 7, it could be seen that there are significant differences in the perception of the ethical dimension of CSR, based on differences in the importance of a sense of belonging. A more pronounced perception of this dimension of CSR could be seen among the respondents who indicated that sense of belonging is important always (M = 4.14), often (M = 4.08) or sometimes (M = 3.72), compared to the respondents who indicated that sense of belonging is never important (M = 1.88).

**Table 7.** Results of ANOVA test of perception for ethical dimension of CSR according to the respondents' IVs.

| Group 1 | Group 2 | Difference | Significance |
|---|---|---|---|
| Sense of belonging (F = 6.844, *p* = 0.000) | | | |
| Never (M = 1.88) | Sometimes (M = 3.72) | (-) 1.84135 | 0.016 |
| Never (M = 1.88) | Often (M = 4.08) | (-) 2.20599 | 0.002 |
| Never (M = 1.88) | Always (M = 4.14) | (-) 2.26199 | 0.001 |
| Sometimes (M = 3.72) | Always (M = 4.14) | (-) 0.42064 | 0.038 |
| Excitement (F = 3.157, *p* = 0.015) | | | |
| Never (M = 3.28) | Often (M = 4.11) | (-) 0.83284 | 0.036 |
| Never (M = 3.28) | Always (M = 4.12) | (-) 0.84167 | 0.030 |
| Warm relationships (F = 6.146, *p* = 0.001) | | | |
| Rarely (M = 3.25) | Always (M = 4.17) | (-) 0.92368 | 0.014 |
| Sometimes (M = 3.65) | Always (M = 4.17) | (-) 0.52028 | 0.004 |
| Being well respected (F = 5.746, *p* = 0.000) | | | |
| Never (M = 2.13) | Sometimes (M = 3.78) | (-) 1.65726 | 0.048 |
| Never (M = 2.13) | Often (M = 4.07) | (-) 1.94722 | 0.010 |
| Never (M = 2.13) | Always (M = 4.04) | (-) 1.91300 | 0.011 |
| Rarely (M = 3.06) | Often (M = 4.07) | (-) 1.00972 | 0.013 |
| Rarely (M = 3.06) | Always (M = 4.04) | (-) 0.97550 | 0.011 |
| Fun and enjoyment in life (F = 6.481, *p* = 0.000) | | | |
| Never (M = 3.11) | Often (M = 4.12) | (-) 1.00330 | 0.002 |
| Never (M = 3.11) | Always (M = 4.11) | (-) 1.00000 | 0.002 |
| Rarely (M = 3.43) | Often (M = 4.12) | (-) 0.68836 | 0.038 |
| Rarely (M = 3.43) | Always (M = 4.11) | (-) 0.68506 | 0.031 |
| Security (F = 6.848, *p* = 0.000) | | | |
| Never (M = 2.38) | Often (M = 4.06) | (-) 1.68981 | 0.001 |
| Never (M = 2.38) | Always (M = 4.07) | (-) 1.69915 | 0.001 |
| Sometimes (M = 3.57) | Always (M = 4.07) | (-) 0.50124 | 0.050 |
| Self-respect (F = 4.320, *p* = 0.002) | | | |
| Never (M = 2.50) | Often (M = 3.99) | (-) 1.49167 | 0.023 |
| Never (M = 2.50) | Always (M = 4.06) | (-) 1.56092 | 0.013 |
| Sense of accomplishment (F = 3.150, *p* = 0.015) | | | |
| Never (M = 2.88) | Always (M = 4.06) | (-) 1.18473 | 0.047 |

Source: research results. Note: white columns below the table fields *Group 1* and the *Group 2* represent the levels of importance of concrete individual values (represented in gray rows above them). Comparisons were made between first two white rows below stated individual values, based on the mean values represented in brackets. Detailed explanation of represented results is provided in the text.

In addition, there are significant differences in the perception of the ethical dimension of CSR based on the excitement IV. Based on the research results, represented in Table 7, it could be noticed that the perception of the ethical dimension of CSR is more pronounced among the respondents who indicated that excitement is always (M = 4.12) or often (M = 4.11) important in life, comparing to the respondents who indicated that this IV is never important (M = 3.28). The research results of the one-factor univariate analysis also indicated significant differences in the perception of the ethical dimension of CSR based on the difference in the importance of warm relationships. Perception of the ethical dimension of CSR is more pronounced among the respondents who indicated that warm relationships are always important (M = 4.17), compared to respondents who indicated that this IV is sometimes (M = 3.65) or rarely (M = 3.25) important.

Numerous significant differences could be noticed in the case of perception of the ethical dimension of CSR based on the importance of the being well respected IV. The

research results, represented in Table 7, indicated that perception of the Ethical dimension of CSR is significantly more pronounced among the respondents who often (M = 4.07), always (M = 4.04), or sometimes (M = 3.78) agree that being well regarded is important than among respondents who indicated that this IV is rarely (M = 3.06) or almost never (M = 2.13) important. Based on the research results, represented in Table 7, it could be seen that perception of the ethical dimension of CSR is more pronounced among the respondents who indicated that a sense of belonging is always important (M = 4.06), compared to the respondents who indicated that this sense is never important (M = 2.88).

For IV, which stands for fun and enjoyment of life, the research results in Table 7 show that differences in the importance of this individual value are reflected in significant differences in the perception of the ethical dimension of CSR. More specifically, perceptions of this CSR dimension are significantly higher among respondents who indicated that fun and enjoyment in life are frequently (M = 4.12) or always (M = 4.11) important to them, compared to respondents who indicated that this IV is important only rarely (M = 3.43) or never (M = 3.11). Perception of the ethical dimension of CSR differs significantly based on the security IV. The research results, represented in Table 7, showed that the perception of the ethical dimension of CSR is significantly more pronounced among respondents who indicated that a sense of security is important always (M = 4.07) or often (M = 4.06) compared to the respondents who indicated that this IV is only sometimes (M = 3.57) or never (M = 2.38) important.

The one-factor univariate analysis also indicated that perception of the ethical dimension of CSR differs based on the IV of self-respect. A more pronounced perception of this dimension of CSR is noticeable among the respondents who indicated that self-respect is important always (M = 4.06) or often (M = 3.99), compared to the respondents who indicated that this IV is never important (M = 2.50).

## 5. Discussion

The authors of this research wanted to look into the relationship between the importance of different IVs among hotel employees and their perceptions of the philanthropic, legal, economic and ethical dimensions of CSR in the business environment in which they work, within the National Park of Kopaonik in Serbia. In addition to the hotels' efforts to influence employees' perceptions and behavior, the findings of this study revealed that the perception of CSR, as a contemporary business principle in our country, differs significantly based on the IVs of employees, which is again correlated with a different degree of importance in IVs of each employee in the hotel. Previous research showed that employees' perception of CSR is positively related to their performance and corporate identification [22], which additionally justified further research of the CSR perception from different perspectives. According to Wong et al. [13], such results contribute to a better understanding of the implementation of CSR in the business environment of developing countries, which is still limited. Previous findings regarding the employees' perceptions of CSR employees in the specific context of Serbia, as a developing post-socialist country, were related to values, but of the national culture. The research results indicated dependence of CSR perception on national culture perception among the employees in the public sector, but not the private companies [23]. Regardless, employees' perceptions of CSR in the context of their individual are still limited. The first step was to identify the dimensions of CSR and assess the importance of basic IVs among hotel employees in a transition country.

In a previous study of IVs in the context of the Serbian tourism industry, Jovanović et al. [57] provided insight into the fact that IVs shape respondents' perceptions of the ethical climate. More precisely, the main findings of the study indicated that managers' perception of the ethical climate is shaped by their sense of belonging, fun and enjoyment of life, warm relationships with others, self-fulfillment and being well respected. On the other hand, in the case of their subordinates, authors found that the perception of ethical climate types is shaped by IVs, such as a sense of belonging and security [57]; interestingly, similar research on the differences in perception of CSR based on IVs was not conducted until

now. Given that the majority of the employees temporarily migrate to Kopaonik to work during the season, it is reasonable to expect that a sense of respect (self-respect and being well respected), security (security) in a new, unfamiliar environment and social fulfillment through close relationships with others are extremely important to them. Furthermore, young people represent the majority of the sample (60.7% of the respondents are 30 or less). Therefore, sense of accomplishment is currently more important to them, compared to fun and enjoyment in life or excitement, because earnings and professional development are often considered the main motives for working within hotels in Kopaonik, as well as the fact that they consider their employment in Kopaonik as a good starting point for further professional development.

Aside from the importance of each IV to hotel employees, the central research question was whether there are significant differences in the perception of the previously determined CSR dimensions (philanthropic, legal, economic and ethical) based on differences in the importance of IVs in the respondents' daily lives. The findings showed that there are indeed significant differences, which is one of the first findings on this topic. Madanaguli et al. [11] stated that the relationship between personality dimensions and CSR perception has never been explored before.

Respondents who showed a stronger orientation towards sense of belonging have a more pronounced perception of philanthropic dimension of CSR. This further indicates that our society would certainly find it easier to adopt business standards of social responsibility if they were presented to the public as desirable or acceptable. In addition, sense of belonging is certainly an important IV, especially for the employees who work seasonally in Kopaonik, primarily because they are distanced from their family and friends. On the other hand, it also allows those who are employed in Kopaonik throughout the whole year to feel included in social interaction, which is often absent during the off-season business.

Additionally, working in the hotel industry is something that implies socializing and direct contact with colleagues, as well as direct contact with guests. Moreover, due to the already mentioned distance between the employees who work seasonally in Kopaonik and their family and friends, these employees are trying to establish closer relationships with their colleagues, as they communicate every day and live together during the season. Respondents who work in hotels throughout the year, on the other hand, have the opportunity to meet a large number of people from various parts of Serbia, with whom they frequently maintain contact even after the tourist season has ended. These results are supported by the study by Heimtun [58] who indicates that tourism provides daily opportunities for social interaction with different people. The philanthropic dimension of CSR also aims to engage organizations in campaigns aimed at helping people in need, which provides an opportunity to engage in activities that are generally different from the usual business activities. Therefore, employees might perceive such activities as fun because they can do something different, especially because it is a business concept that is not yet sufficiently developed in Serbia, but on the other hand, as pleasure because of the opportunity to provide assistance. All of the abovementioned facts might increase the sense of security of the employees, primarily because they are also a part of the community. Moreover, donations and volunteer activities could create the sense that an organization that takes care of others will also take care of its employees.

A person with an increased sense of self-respect is someone who values other people and who is also valued by other people [55,59]. Therefore, it is not surprising that such people have a desire to help others through philanthropic activities. This finding implies that participating in philanthropic activities would boost employees' sense of accomplishment, or more precisely, be based on carrying out good deeds.

Regarding the perception of Legal dimension of CSR, results of the one-factor univariate analysis showed significant differences in perception of this dimension only in the case of differences in the evaluation of individuals' self-respect. According to the study conducted by Basta [60], an individual without Self-respect easily gives up on moral values, which are the basis of law and legal acts, as well as the essence of Legal dimension of CSR.

The research results also show that there are significant differences in the perception of the economic dimension of CSR, based on six out of the nine examined IVs. Employees might perceive that their better business results will strengthen their business position, which might also increase their sense of belonging to a particular organization, which usually strives toward retaining productive employees. If the organization is economically successful, salaries are paid regularly and as expected, and employees are reassured, which means they can enjoy their free time, free from existential problems. An important aspect of the economic dimension of CSR is the establishment of a long-term strategy for economic growth, which could further help to reduce the problems of employees related to the existential needs already mentioned. As a result, employees may see the value of their business tasks in meeting the organization's broader goals, which may contribute to making employees feel valued in the workplace. Furthermore, working in a successful organization with high productivity (as part of the economic dimension of CSR) is directly related to a sense of accomplishment, as our findings show.

When it comes to the last dimension, the research results showed that significant differences in a perception of the ethical dimension of CSR exist in the case of differences in eight out of nine IVs. Only the IV of self-fulfillment did not show a significant difference in the perception of the ethical dimension of CSR, which means that those with more or less importance of self-fulfillment in life will also behave according to the ethical issues. This could be interpreted to mean that both would be able, for example, to consciously portray a product/service as better than it is. As ethics deals with issues that are considered acceptable and unacceptable in society, it is logical that someone who indicated the sense of belonging as important will have a more pronounced perception of the ethical dimension of CSR.

All aforementioned findings contribute to a better understanding of the micro-level consequences of perceiving CSR, based on individual outcomes in a transitional business environment. The main contribution is reflected in the fact that such studies are still limited, as already highlighted by Zhang et al. [24].

## 6. Conclusions

Because several IVs describe a relationship with others, and because humans are social, collective beings, the authors believe that there is a relationship between the moral values of the hotel managers interviewed and the perception of the philanthropic, legal, economic and ethical dimensions of CSR within the organization by all employees. It is important to consider all of these findings, especially given the environment in which these hotels operate. This is characterized not only by the fact that it is a protected natural asset, but also by the fact that a small area becomes a central meeting point during the winter season, not only for tourists, but also for a large number of employees from all over the country as well as from abroad. This is the case with many hotels around the world, so the importance of this research is even greater.

The main findings of this research contribute to a better understanding of implementing the concept of CSR in the business environment of dominating small and medium hotels within developing countries, which is still limited. Moreover, the contribution of this research is additionally reflected in the fact that it is focused on detailed explanation of personality dimensions, such as individual values, in the context of shaping the CSR initiatives, reflecting the relationships that have not been studied before. Aside from the research findings contributing to a better understanding of differences in CSR perceptions, based on differences in the hotel employees' IVs within the transitional environment of small and medium hotels, this study contains several limitations. First, this research was focused only on hotel employees within the tourism sector. Further research might be focused on dividing the sample on a various basis, for example by the respondents' socio-demographic characteristics, their employment position (managerial or subordinating) or a sector division. Such findings would provide additional insight into the CSR perceptions. Furthermore, the research was conducted within one country, while similar studies

might be conducted within other transitional countries and sectors, which would provide a significant basis of information to compare.

*Practical Implications*

Besides the theoretical contribution, the main findings of this research provide the basis of information for practical implementation in the hotel industry on how IVs can motivate employees to understand and participate in the proactive management of CSR activities within sensitive environments, such as national parks or other protected areas that become the central gathering place of both tourists and employees. In addition to improvement in business performances, such business practice might also create a desirable business environment, in accordance with employees' preferable individual values. Finally, the main research findings provide new insights that might inform hotels in their planning and execution of CSR communication aimed at their employees, focusing on individual values instead of on the society-centred values.

**Author Contributions:** Conceptualization, M.M.L.; Methodology, M.M.L., T.J., O.M.I. and M.D.V.; Software, A.S.; Validation, K.K. and M.N.K.; Formal analysis, T.J., S.T. and M.D.V.; Investigation, M.M.L., A.I.D. and J.T.; Resources, A.S.D., A.I.D. and Z.Z.; Data curation, A.S.D. and V.S.; Writing—original draft, M.S., V.S., A.I.D. and I.B.; Writing—review & editing, K.K., M.D.V., Ž.A. and Z.Z.; Visualization, S.T. and J.T.; Supervision, T.J., O.M.I., M.N.K. and A.S.; Project administration, A.S.D., M.S. and Ž.A.; Funding acquisition, M.M.L. and I.B. All authors have read and agreed to the published version of the manuscript.

**Funding:** This research received no external funding.

**Institutional Review Board Statement:** Ethical review and approval were waived for this study due to the fact, according to legal requirements within researched country, it was not obligatory element. The consent of the research participants was sufficient for conducting the study.

**Informed Consent Statement:** Informed consent was obtained from all subjects involved in the study.

**Data Availability Statement:** Data basis is not available via public storage.

**Acknowledgments:** The methodology used in this paper represents a basis for further research development as a part of the project approved by the Autonomous Province of Vojvodina, Provincial Secretariat for Higher Education and Scientific-Research Activity, Program 0201, with of the project title "Research of the entrepreneurial potentials among the local population for using the thermomineral water resources of Vojvodina", registration number: 142-451-3137/2022-04 (2021–2024).

**Conflicts of Interest:** The authors declare no conflict of interest.

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
