# Peer review of "Tourism Stakeholder Perspectives on Corporate Social Responsibility in Serbia: The Perception of Hotel Employees"

_sustainability, doi:10.3390/su15054010_

Round 1

Reviewer 1 Report (Previous Reviewer 1)

Some structural changes were made, and paper is improved. Still, some changes and upgrades are needed to be done.

I still believe that the part that includes the Google scholar link related to Carroll’s research (Introduction) should be re-written. This data is continuously changing, so you are stating something that probably has been changed in the last few weeks. Even posting a link is not usual approach while writing an academic paper, but if you think it is essential, you should at least make an appropriate statement that will be correct whenever the reader open your article.

This research is focused on the hotel industry in a developing country, such as Serbia… it is not such as but concretely in Serbia… Re-phrase: This research is focused on the hotel in Serbia, as an example of developing countries

In the methodology section, you have mention that the method used for the analysis is ANOVA. In the findings, you have started presentation of findings with Kaiser-Meyer-Olkin’s indicator, which (as per my knowledge) is part of factorial analysis. You should elaborate this in the methodology.

In the methodology, you should clearly state what are your variables and description of the variables. You dis it in descriptive way in 3.1. but it is very confusing while reading. A table view with clear presentation is appreciated.

Author Response

Please,

find in attach an answer.

Reviewer 2 Report (Previous Reviewer 3)

The article titled “Perspective Of The Corporate Social Responsibility Of Tourism Stakeholders In Developing Countries: Perception Of Hotel Employees” sets out to examine the viewpoints of Serbian respondents with regard to CSR policies in hotels.

While the topic is interesting, considering the current business environment that requires the addition on a CSR components, the paper has many issues with the understanding of the research premises. The following evaluation points apply for this manuscript:

1. The study's goals should be defined.  The goals of the research are frequently stated at the conclusion of the introduction.

2. The text reflects many issues with understanding the flow of ideas. 

For instance, in the introduction, the paragraph ending on row 61 discussed CSR, then the following paragraph (starting on row 62) states: “These hotels have the necessary infrastructure in terms of modern technologies, 62 capital, and knowledge to carry out organizational social responsibility. In recent years, 63 hotel chain branches in Serbia have implemented a variety of social responsibility 64 programs to assist in the resolution of environmental and community issues. In addition 65 to addressing significant social issues, these hotels have improved their relationships with 66 customers, employees, and suppliersThis is reflected in how the hotel is perceived, as well 67 as in the stakeholders' loyalty and trust.” 

Why are the authors referencing ‘These hotels’? What is the link to the previous idea? Why aren’t any references provided for the explained context of Serbia’s switch towards CSR?

Then, in the same paragraph the authors continue to address CSR :” Adopting the wider concept of CSR, Carroll [12] 68 argued that CSR needs to include four types of responsibilities: economic, legal, ethical 69 and philanthropic.”

The ideas presented should be logical. 

Similar issues with understanding the context explained appear in different parts of the manuscript or overcomplicated phrases that are difficult to understand (e.g.  rows 103-105, 124-135, 146-149, and so on).

3. Error with citation on row 81. 

4. In the introduction, the Authors state: “Concerning that, the main objective of this research is 93 focused on providing a unifying framework highlighting the importance of individual 94 values in shaping the CSR perception of employees in the context of the hotel industry. 95” How is this objective possible giving the limited analysis focused on ANOVA? It seems that this paragraph lacks consistency with the actual framework of the research.

5. In its current form the Introduction does not provide a clear understanding of the intended audience, the motivation for the research, and the novelty and relevance of the paper.

6. “A thorough review of the literature suggests that the personal characteristics of 162 employees may also affect attitudes toward CSR.” Which studies are included in this “thorough review”?

7. Why are there two sections 2.2. Individual Values?

8. Authors may find the following reference useful: Bibi, S., Khan, A., Hayat, H., Panniello, U., Alam, M., & Farid, T. (2022). Do hotel employees really care for corporate social responsibility (CSR): A happiness approach to employee innovativeness. Current Issues in Tourism, 25(4), 541-558.

9. The study’s hypotheses are not clearly presented based on the literature review. What is the purpose of the paper? 

10. Considering the data collection process (2013-2018), how can the authors offer support for the novelty of the study, especially on a prominent topic such as CSR? Why is the research relevant in this context?

11. For the sake of research transparency, the scale items’ sources need to be provided in Table II.

12. The other scale items for Individual values items should also be presented (with their associated sources).

13. Similar to Table II Factor analysis for CSR, similar details are necessary to IVs.

14. The Groups shown in the tables are not properly explained by the Authors. How were these groups established? Why are they relevant? How many respondents were included in each group for every analysis?

15. The manuscript does not offer compelling arguments for the relevancy of the analysis. The manuscript repeatedly mentions “significant differences”. However, focusing to a large extent on the average values of different dimensions, the manuscript offers a limited perspective on the analysis. 

16. The manuscript should consider additional perspectives of expanding the research: correlation, segmentation of ANOVA, MANOVA, regression, hierarchical regression, interactions…  

17. The Discussion section only reflects reiterations of the results’ interpretation. The manuscript lacks in comparing the results with relevant research.

18. In the text, the Authors mention: “There is also a clear connection”, “The authors of this research wanted to investigate the relationship”. However, these interpretations are not exactly accurate when focusing solely on one indicator of central tendency.

19. The authors mention: “The average value of Fun and enjoyment in life is more pronounced among the 624 younger ones in our sample and it decreases in parallel with the increase in the age 625 structure of the respondents.” Which analysis result highlighted this result?

20. The authors mention: “Respect for others, respect for rights and similar issues are moral, or more precisely 635 ethical elements, so it is clear in terms of Self-respect that it is correlated with the Ethical 636 dimension of CSR.” Which analysis result highlighted this result?

To a large extent, the Authors propose conclusions that highlight relationships or correlations, however the analysis did not focus on any of these data analysis techniques. Thus, many interpretations lack accuracy.

21. Moreover, the practical implications should be included in a separate section.

22. I advise improving the English language by a native speaker, to adjust the overly complicated phrases and sentences. 

As a conclusion, there are many structural issues with the manuscript. Overall, there are very few original contributions of the study, few recent studies used to highlight the context of the paper, also the hypotheses of the manuscript are missing.  Most importantly, major issues with the paper are regarding the topicality and the relevancy of a study with an older data set (according to the data collection process). 

Author Response

Please,

find in attach an answer.

Reviewer 3 Report (New Reviewer)

Thank you for giving me an opportunity to review the paper entitled: “Environmental degradation and climate change bound tourism exercises: alternative ecotourism management perspectives within the rural conservation sites and farming communities”.

There are several suggestions given for the improvement of the paper:

1.     The abstract is not well written. The current form abstract is not well written. At first, author could talk about the main purpose of the study and then novelty of the study. Then talk about the methodology such as sample and software used to analyse the model and hypotheses. After that, author could include the key findings and contribution. I do not see any sentences talk about the method used.

2.     Overall, I think the introduction is not well written. There are lots of grammar issues, syntax, sentence structure, etc. Apart from this, the intention of the study is not well explained and not clearly justified. I found it difficult to understand the introduction.

3.     The gap of the study is not well explained. Should focus on the topic of this study and argue why responsible tourism are discussed in this study and what’s new about this study.

4.     I believe that there are several similar studies which can be discussed in the introduction. What make this study unique, and novel as compared to past studies? The importance of the study is not sufficiently explained. Author should relate the gap to the issue.

5.     The sequences of the introduction element are required to be reorganized. The introduction should be included as follows:

(1) Briefly describe and illustrate the current issue.,

(2) Why such study with proposed research gaps is important?,

(3) How this research gap relates to current issue?,

(4) Why such underexplored piece of work is important to be tested in your study?,

(5) Any similar studies conducted in the past?,

(6) What is the uniqueness of this study as compared with past empirical studies? and

(7) What are your research objectives?

(8) What are the contributions of the studies?

6.     Should re-structure the research methodology into two main sections: (1) sampling and research procedure and (2) research instruments.

7.     Please clearly define the target respondents. Any criteria to select the respondents? How to consider them as respondents?

8.     What was sampling technique used to select respondent? Why? How? Next question is how do you ensure the generalizability and representativeness of the sample toward the targeted population? Any selection criteria? How do you select the respondent for your study? Any procedure of selection? Please justify.

9.     The procedure of data collection is not clear. Should provide more information about how authors collect the data, how to approach the respondent, how to identify them to participate in the survey? Try not to exaggerate it and the explanation should be more reasonable and logic.

10.  There is limited demographic information about the sample collected. Please provide more details.

11.  What software used to analyse the data? Should provide explanation as well.

12.  The discussion and conclusion section structure should be revised as follows:

Discussion of key findings

Theoretical Implications

Practical/Managerial Implications

Limitations and Future Research

Conclusion

13.  For discussion, authors need to ensure the key findings are discussed. The discussion section is where you delve into the meaning, importance and relevance of your results. It should focus on explaining and evaluating what you found, showing how it relates to your literature review and research questions, and making an argument in support of your overall conclusion.

14.  Should have a standalone section for implications. How do you imply these findings? I would suggest author to provide implications based on the current practices and policies.

15.  Should have a standalone section for limitations and future research recommendation. Suggest author to carefully identify potential weakness of this study and propose suggestion for future research.

Author Response

Please,

find in attach an answer.

Round 2

Reviewer 2 Report (Previous Reviewer 3)

Based on the revised version of the paper, the following recommendations apply:

1. As previously mentioned, the paper does not offer a wide contextualization of the topic, using recent and relevant studies on the topic of the article.

2. Perhaps, the Authors could include short descriptions of the Groups below the tables (in a Note related to the table) to increase the readability factor of the paper.

3. In the conclusions’ section, the manuscript should highlight the original/theoretical contribution of the study. In its current form, certain contributions are mentioned in the discussion (e.g. “such results contribute to a better understanding of the 473 implementation of CSR in the business environment of developing countries, which is still 474 limited”)

4. The discussion of the results  (Section 5) is extensive, however there are limited comparisons to existing studies on employee perception on CSR practices. The Authors should focus more on comparing and contrasting the results with different studies on this topic, and not just reiterate the explanations of the results mentioned in Section 4. This aspect was provided as a recommendation in my previous review.

Author Response

Thank you so much for all your suggestions.

Authors

Reviewer 3 Report (New Reviewer)

I am happy to review the improved version of the manuscript. The manuscript is well revised. Thus, no further comments from me. 

Author Response

Thank you so much for all your suggestions.

Authors. 

Round 3

Reviewer 2 Report (Previous Reviewer 3)

In the current format, the Authors do not seem to address the issue of academic writing. As previously mentioned, the Discussion section only reflects reiterations of the results’ interpretation. Thus, the paper is not concisely presented. This aspect extends to the Literature review.

In their response, the Authors state: “Previous literature regarding the research topic of the paper is rather limited”. Obviously, there are multiple studies on the topic of CSR practices from the perspective of employee perception developed other countries, as well as in Serbia. A more rigorous examination of existing literature is recommended for a comprehensive manuscript. 

Recommended reference: Radovanović, G., Miletić, L., Karović, S., Mijatović, M. D., & Bubulj, M. (2022). Influence of National Culture in Companies with Different Ownership on Employees’ CSR Perception in a Developing Country: The Case of Serbia. Sustainability, 14(4), 2226.

As previously recommended, for research transparency, the scale items’ sources need to be provided for IVs, as well (with their sources).

Study Limitations should be clearly presented, as there are many aspects that apply in this case. Briefly, the paper mentions future directions, without fully addressing the issues of the study. E.g. from text: “this study contains several 672 limitations. First of all, further research might be focused on dividing the sample on a various basis” (+ English language error)

Author Response

Please find in attach Response.

And thank you!

Authors

Round 4

Reviewer 2 Report (Previous Reviewer 3)

In the current format, the Authors do not seem to address the issue of academic writing. As previously mentioned, the Discussion section only reflects reiterations of the results’ interpretation.  Sections 4 and 5 seem to reflect similar elements. Thus, the discussion section could be shortened to reflect concise ideas. 

Author Response

Please,

find in attach file. 

This manuscript is a resubmission of an earlier submission. The following is a list of the peer review reports and author responses from that submission.

Round 1

Reviewer 1 Report

Carroll’s Pyramid and his contribution to the topic are absolutely not questionable. But, in terms of your paper and research topic, this is not relevant especially putting the external link to the academic paper. Your research aim is about CSR issues in the hotel industry in the example of Serbia. Thus, you should re-adjust the introduction to start with the general issues but somehow address the topics. Somehow you started elaborating on very general issues in the introduction. E.g. you can use the text from 83 -103 text to introduce the readers to what will be your topic… I am not saying to re-shift but to reuse for the introduction to the topic and your paper.

You mentioned data collection started in 2013. Which period is covered by your research?  Previously you mentioned that you used the questionnaire for data collection. Can you elaborate? You should identify and elaborate more on your variables. The empirical model should be elaborated on and how you did your analyses. Research methodology is very weak and it must be allocated more.

The paper has the potential and the topic is very interesting. You should revised the paper to improve the clarity and to provide better structure. 

The discussion section must be included. This is the most important part to address the novelty in the literature and compare your findings with the existing literature.

The conclusion included some parts that should not be there. The conclusion is to conclude the paper and highlight the main findings. However, you included there some parts of how you did your research (e.g. 493-497). What were your limitations and proposals for further researches?

Author Response

Please,

find in attach response.

Thank You for all your comments. 

Reviewer 2 Report

English usage in this manuscript needs extensive revision for language and grammar, and it must be substantially improved (i.e., line 66, line 180 etc.)

Pease rewrite the Abstract in a more coherent and accurate way.  You can display briefly the aim, the gist, the innovative of the manuscript.

Paragraph 2.1: Authors refer in the title “Instrument, Data Collection Procedure and Statistical Analysis”, however they did not display any information about the analysis they applied. Please explain further.

Α limitation is that the survey was conducted in 2013. Is it safe to be published after 8 years?? There are no changes on influencing factors and on the results are presented.

Paragraph 2.2: It will be clearer if the information of the sample presented in a Table.

Section3: Authors show some results, but it is not explained how they came out.

The statistical analysis was conducted seems to be very basic and simple descriptive. Applying more advanced statistical techniques will help you to answer thoroughly the research questions of the survey and improve the quality of the manuscript.

Author Response

(The authors gave the same response as above.)

Reviewer 3 Report

The manuscript addresses the concept of CSR in the tourism industry. Nonetheless, the paper can be improved in various sections.
Considering the examined manuscript, I recommend the improvement of the following aspects:
1.    The primary original features of the work should be highlighted in the abstract.
2.    The paper’s introduction should be strengthened to emphasize its importance in relation to existing research gaps that the present study aims to examine.
3.    The objectives of the study should be more clearly explained.
4.    Additionally, the introduction mentions a limited set of recent sources. Especially given the broad examination of the CSR topic, more recent sources can be highlighted to create relevant parallels for the Serbian context of the study (and the context of the journal).
5.    The Literature Review should be separated from the Introduction.
6.    As previously mentioned in relation to the introduction section, additional current resources should be included to the context of the study for the literature review. Authors may find useful the following papers:
•    Madanaguli, A., Srivastava, S., Ferraris, A., & Dhir, A. (2022). Corporate social responsibility and sustainability in the tourism sector: A systematic literature review and future outlook. Sustainable Development, 30(3), 447-461.
•    Wong, A. K. F., Kim, S., & Lee, S. (2022). The evolution, progress, and the future of corporate social responsibility: Comprehensive review of hospitality and tourism articles. International Journal of Hospitality & Tourism Administration, 23(1), 1-33.
•    Zhang, J., Xie, C., & Morrison, A. M. (2021). The effect of corporate social responsibility on hotel employee safety behavior during COVID-19: The moderation of belief restoration and negative emotions. Journal of Hospitality and Tourism Management, 46, 233-243.
7.    The Literature Review could benefit of certain restructuring of ideas. For instance, on rows 143-145 the authors mention the form of measurement of certain variables.
8.    The section titled ‘Research Methodology’ needs major improvements in terms of explaining the data collection process.
9.    The Authors mention ‘The research results are part of a broader study on business ethics and CSR. Data collection for this study began in 2013 (rows 173-174)’. How can the Authors support the original and relevant aspects of the research considering this 2013 timeframe?
10.    The scale items used for this research are not presented for research transparency.
11.    On rows 174-175, an idea related to a factor analysis is presented. However, the format, results and the accuracy of the analysis are not presented.
12.    The section titled ‘Results and Discussion’ starts too abruptly without presenting the analyses that are developed to account for the research objectives. The Authors do not offer clear explanations on the Groups reflected in the tables and the reasoning behind them.
13.    The analysis is very simple. The authors should consider new ways of expanding the research and providing a more thorough analysis (additional tests could add more value to the paper: normality tests, correlation, segmentation of ANOVA, MANOVA…)
14.    In the text, the Authors mention: “There is also a clear connection”, “The authors of this research wanted to investigate the relationship”. However, these interpretations are not exactly accurate when focusing solely on one indicator of central tendency and F-tests.
15.    Moreover, in discussing the results, the findings should be compared to previous studies’ outcomes.
16.    In its current form, the Conclusion section reiterates the results. However, the original contributions of the study need to be addressed (in relation to clearly established research objectives).
17.    The Managerial implications of the study need to be highlighted.
18.    Also, the limitations of the study need to be transparently presented.

Author Response

(The authors gave the same response as above.)

Round 2

Reviewer 1 Report

The paper is improved, especially the structure.

The literature review should be a bigger section than the introduction. Think of re-allocate some parts from there if it is possible.

Please review the grammar. Usually, you should use the past tense for stating the literature review. In some cases, you used present continuous (e.g. Madanaguli et al. [11] are indicating…). Be consistent in this regard.

Reviewer 2 Report

The authors have made a sincere effort to improve the text. However, there are points, mainly methodological, that have not been improved.

Particularly, regarding the statistical analysis of the results, it remains almost insufficient. The honest answer of the authors that "Within previously published papers, we conducted various analyses, such as factor analysis, GLM, etc. We did not put these results in this study, in order to avoid a duplication of published results ", indicates that the significant results of this valuable research have already been published. Therefore, I would suggest to avoid duplicate publication of the same material.

Reviewer 3 Report

The manuscript has been improved in various sections and the new version showcases additional contributions to CSR research. The explanations for the data collection process and the chosen analysis techniques are appreciated, however, the analysis is still simplistic in nature. Moreover, it is still uncommon to mention an analysis (factor analysis) in an academic paper, but not showcase the results (because they were published in previous papers).

1. In the Introduction, a final paragraph should be included to showcase the structure of the upcoming sections of the manuscript.

2. In the new format, the Introduction appear longer than the Literature review.

3. I would recommend expanding the Individual values with definitions of the concepts.

4. I would still recommend the adding of the factor analysis results, because the analysis should showcase reliability, accuracy and transparency. Readership should be informed of the results and all relevant results should be transparently presented. 

5. Especially in the results’ section, the article seems too long, due to reiteration of the results. The results could be summarized in a more relevant and interesting manner.

6. Moreover, the results are not compared to existing studies. 

7. Below the tables, the Authors could provide notes on the differences between Groups 1 and 2.

8. From my previous review, point 13 has not been taken into consideration in expanding the research. Descriptive statistics could be included in table 2. Additionally, Levene’s test or other tests of normality could be included (Kolmogorov-Smirnov and Shapiro-Wilk) to enhance the understanding of ANOVA.

Best of luck to the Authors with their research opportunities!